# Role of Structural and Non-Structural Proteins and Therapeutic Targets of SARS-CoV-2 for COVID-19

**DOI:** 10.3390/cells10040821

**Published:** 2021-04-06

**Authors:** Rohitash Yadav, Jitendra Kumar Chaudhary, Neeraj Jain, Pankaj Kumar Chaudhary, Supriya Khanra, Puneet Dhamija, Ambika Sharma, Ashish Kumar, Shailendra Handu

**Affiliations:** 1Department of Pharmacology, All India Institute of Medical Sciences (AIIMS), Rishikesh 249203, India; drpdhamija@gmail.com (P.D.); shailendra.handu@gmail.com (S.H.); 2Department of Zoology, Shivaji College, University of Delhi, Delhi 110027, India; jnujitendra@gmail.com; 3Department of Medical Oncology & Hematology, All India Institute of Medical Sciences (AIIMS), Rishikesh 249203, India; neeraj.monc@aiimsrishikesh.edu.in; 4Molecular Biology & Proteomics Laboratory, Department of Biotechnology, Indian Institute of Technology (IIT), Roorkee 247667, India; pkchaudharydu@gmail.com; 5Uttaranchal Institute of Pharmaceutical Sciences, Dehradun 248007, India; supriyakhanra05@gmail.com; 6Department of Biochemistry, U.P. Pt. Deen Dayal Upadhyaya Veterinary Science University, Mathura 281001, India; ambikavet78@gmail.com; 7Department of Biochemistry, All India Institute of Medical Sciences, Rishikesh 249203, India; ashishvet77@gmail.com

**Keywords:** coronavirus, structural proteins, Angiotensin converting enzyme 2, Coronavirus disease-19, SARS-CoV-2

## Abstract

Coronavirus belongs to the family of Coronaviridae, comprising single-stranded, positive-sense RNA genome (+ ssRNA) of around 26 to 32 kilobases, and has been known to cause infection to a myriad of mammalian hosts, such as humans, cats, bats, civets, dogs, and camels with varied consequences in terms of death and debilitation. Strikingly, novel coronavirus (2019-nCoV), later renamed as severe acute respiratory syndrome coronavirus-2 (SARS-CoV-2), and found to be the causative agent of coronavirus disease-19 (COVID-19), shows 88% of sequence identity with bat-SL-CoVZC45 and bat-SL-CoVZXC21, 79% with SARS-CoV and 50% with MERS-CoV, respectively. Despite key amino acid residual variability, there is an incredible structural similarity between the receptor binding domain (RBD) of spike protein (S) of SARS-CoV-2 and SARS-CoV. During infection, spike protein of SARS-CoV-2 compared to SARS-CoV displays 10–20 times greater affinity for its cognate host cell receptor, angiotensin-converting enzyme 2 (ACE2), leading proteolytic cleavage of S protein by transmembrane protease serine 2 (TMPRSS2). Following cellular entry, the ORF-1a and ORF-1ab, located downstream to 5′ end of + ssRNA genome, undergo translation, thereby forming two large polyproteins, pp1a and pp1ab. These polyproteins, following protease-induced cleavage and molecular assembly, form functional viral RNA polymerase, also referred to as replicase. Thereafter, uninterrupted orchestrated replication-transcription molecular events lead to the synthesis of multiple nested sets of subgenomic mRNAs (sgRNAs), which are finally translated to several structural and accessory proteins participating in structure formation and various molecular functions of virus, respectively. These multiple structural proteins assemble and encapsulate genomic RNA (gRNA), resulting in numerous viral progenies, which eventually exit the host cell, and spread infection to rest of the body. In this review, we primarily focus on genomic organization, structural and non-structural protein components, and potential prospective molecular targets for development of therapeutic drugs, convalescent plasm therapy, and a myriad of potential vaccines to tackle SARS-CoV-2 infection.

## 1. Introduction

In late December 2019, an acute case of respiratory diseases was reported in Wuhan, Hubei People’s Republic of China. Initially, clinical symptoms such as fever, sore throat, and respiratory distress amongst others overlapped with viral pneumonia. However, after genomic analysis of respiratory samples from patients, it was revealed as novel coronavirus (2019-nCoV) pneumonia [1]. Later on, 2019-nCoV was finally renamed as SARS-CoV-2 by the International Committee on Taxonomy of Viruses (ICTV) owing to its genetic similarity to an earlier known coronavirus (SARS-CoV) [2]. This coronavirus-induced disease was officially announced as Coronavirus Disease-19 (COVID-19), and on 11 March 2020, World Health Organization (WHO) declared it as a pandemic owing to its vast magnitude of global spread. As of 30 March, 2021, nearly 128 million of confirmed cases and more than 2.8 million of deaths has been reported across the globe (worldometers.info/coronavirus; accessed on 30th March 2021). Coronavirus is a large family of viruses that are capable of causing moderate illness such as common cold, to more severe life-threatening diseases like acute respiratory distress syndrome (ARDS) and organ malfunctions [3]. SARS-CoV-2 is taxonomically placed under order *Nidovirales*, subfamily *Orthocoronavirinae*, and four genera namely *Alphacoronavirus* (α), *Betacoronavirus* (β), *Gammacoronavirus* (γ), and *Deltacoronavirus* (δ) [4]. All viruses of *Nidovirales* order share several common features, such as they possess conserved genomic organization, replicase gene located downstream to 5′-UTR, expression of downstream genes following synthesis of multiple 3′-nested subgenomic mRNAs, expression of genes by ribosomal frameshift mechanism, and multiple unique intrinsic enzymatic activities in the large replicase-transcriptase polypeptide among others [5]. People are susceptible to infection by seven of these viruses, including 229E (α), NL63 (α), OC43 (β), HKU1 (β), MERS-CoV (β), SARS-CoV (β), and SARS-CoV-2 (β) [6]. Among them, the four HCoVs, namely, 229E, NL63, HKU1, and OC43, cause very mild to moderate upper-respiratory tract illness, whereas SARS-CoV, MERS-CoV, and SARS-CoV-2 are quite deadly, and may cause fatal respiratory disease involving lower respiratory tract and down right to alveoli of lungs [6]. SARS-CoV-2 shows 88% sequence identity with bat-SL-CoVZC45 and bat-SL-CoVZXC21, and 79% and around 50% sequence identity with SARS-CoV and MERS-CoV, respectively. Therefore, SARS-CoV-2 is phylogenetically closer to bat-related SARS-CoV vis-à-vis MERS-related CoV and SARS-related CoV, indicating that it might have evolved from bats [7]. This knowledge of similarities and differences of SARS-CoV-2 with viruses, which are known to have caused earlier outbreaks, may help in developing better understanding of etiology, and consequently design curative strategy to tackle current pandemic. In contrast to SARS-CoV-2, MERS-CoV infection was first reported in Jordan, Saudi Arabia in 2012 [8]. Humans contract MERS-CoV following coming in contact with infected camel as well as infected humans. Since 2012, 27 countries have reported more than 24 thousand MERS cases, of which the majority of cases have occurred in Saudi Arabia. SARS-CoV was found in China in 2003 [9], and has originated in bats, and was amplified in other animal reservoir before infecting humans. The outbreak was contained in mid-2003 with the implementation of infection control practices such as isolation and quarantine. Since then, a handful of cases have occurred due to laboratory accidents and related exposure which have been comprehensively reviewed [10]. SARS-CoV-2 is a new strain of corona virus that has not been traced in humans previously. In this review, we primarily focus on multiple features of SARS-CoV-2, including genomic organization, structural and non-structural protein components, and potential prospective molecular targets, against which either various FDA approved drugs used for other diseaseare being repurposed, or novel drugs under various phases of clinical trial are being tested. In addition, we also focus on plasma therapy and antibody cocktail, as well as 13 approved/authorized vaccines being administered worldwide to tackle current pandemic.

## 2. Genomic Arrangement of SARS-CoV-2

The nucleotide length of genome of SARS-CoV-2 is ranging from 26.0 kb to 32.0 kb, having variable number of open reading frame (ORFs), and its genomic organization is considerably similar to other known HCoVs [11]. For instance, coronaviruses characterized till date are enveloped viruses with numerous surface-projected club-like spikes, and they possess an unsegmented, single-stranded (ss), +ve sense RNA genome with 5′-cap and 3′-poly(A) tail, allowing it to act as functional mRNA for translational synthesis of the replicase polyproteins [5]. Two-thirds of the viral genomic region i.e., initial 20 kb lying downstream to 5′-end, occupies replicase gene referred to as Open Reading Frame 1a and ab (ORF1ab), which potentially encode the nonstructural proteins (nsps) referred to as pp1a and pp1ab polyproteins, respectively. The pp1a non-structural protein corresponds to NSP1 to NSP11 and pp1ab non-structural protein comprises of NSP12 to NSP16. The remaining 10 kb region preceding 3′-end encodes various structural proteins involving surface (S), envelope (E), membrane (M), and nucleocapsid (N) proteins. Additionally, the structural genes encode nine accessory proteins, encoded by ORF3a, ORF3d, ORF6, ORF7a, ORF7b, ORF8, ORF9b, ORF14, and ORF10 genes [12]. Furthermore, the genomic region lying immediate to 5′-end possesses two distinct domains, leader sequence and untranslated region (UTR), capable of forming multitude of stem loop structures prerequisite for replication and transcription of viral genome. The transcriptional regulatory sequences (TRSs) precede each structural and accessory gene which are prerequisite for their functional expression. The structural component of 3′-UTR is required for replication of viral RNA (Figure 1) [5].

## 3. Structural Organization/Assembly of SARS-CoV-2

The SARS-CoV-2 are spherical, enveloped, around 80–120 nm in diameter, with multiple outwardly projected club-like homotrimeric, glycosylated S proteins imparting them incredible appearance of a solar corona, prompting their popular name, CoVs. Enclosed within the lipid bilayer envelope of the virion is helically symmetrical nucleocapsids comprising complex of +ssRNA and capsid proteins. There are four important structural proteins—spike (S), membrane (M), envelope (E), and nucleocapsid (N) proteins—that are encoded by structural genes located within the region preceeding 3′ end of genome. Apart from above-mentioned structural proteins, there are several non-structural and/or accessory proteins, which together are responsible for the structural and functional aspects of virus. The various structural and functional aspects of aforementioned proteins are summarized below.

## 4. Description of Structural Proteins

The S protein of virion is a type I transmembrane N-linked glycosylated protein (150–200 kDa) consisting of 1273 amino acids, and displays varying degree of conservation across the *Coronaviridae* family. Following synthesis, three polypeptide chains of S protein associate with each other, forming homotrimeric assemblage. Each monomeric S protein consists of S1 and S2 subunits, which together possess multiple functional domains, from N to C-termini as follow: N-terminal domain (NTD), receptor binding motif (RBM) containing receptor binding domain (RBD), furin cleavage site (S1/S2, which is likely to be cleaved by the TMPRSS2 protease), fusion peptide (FP), central helix (CH), connecting domain (CD), heptad repeat (HR1/2) domain, transmembrane domain (TM), and cytoplasmic tail (CT). Furthermore, each monomeric S protein folds into a three-dimensional structure with three distict topological domains—i.e., head, stalk, and cytoplasmic tail [13]. The RBD/S1 (~200 amino acids) of S protein undergoes down-to-up conformational transition during interaction with membrane bound ACE2 recepetor, facilitating cell recognition and binding. In addition, RBD substantially contributes towards the overall mechanical stability of homotrimeric spike [14]. The S2 subunit with its fusion peptide (FP), central helix (CH), connecting domain (CD), heptad repeat (HR1/2) domain mediates integration between viral and host cell membrane. Evetually, the interaction between receptor binding motif of RBD of S1 subunit and ACE2 receptor leads to entry into host cells. Another remarkable feature of the S protein is its substantial N-linked glycosylation covering large proportion of protein’s surface area, and shows conformation-dependent dyanamic changes. Apart from shielding, *N*-glycans especially at N165 and N234 positions are involved in modulation of RBD conformation, which may be used as potential therapeutic target [13]. Besides, the S protein consists of disulfide bonded extracellular N-terminus, a transmembrane domain, and a short intracellular C-terminal part with palmitoylation [15]. The S protein plays major determinant of host immune response, and is involved in viral pathogenesis through activation of endoplasmic reticulum (ER) stress response [15,16], and therefore any mutational change may lead to altered pathogenesis.

The membrane (M) protein is O-linked glycoprotein of around 25–30 kDa, and is most abundant amongst various structural proteins, and possesses three distinct transmembrane domains [17]. The homodimeric M protein associates with other viral structural proteins, including nucleocapsid, facilitating the molecular assembly of virus particles as well as may be involved during pathogenesis [18]. Although M protein possesses diverse amino acid composition, but it is structurally preserved across the various genera [19]. Except β-CoVs and δ-CoVs which shows O-linked glycosylation, other coronavirus M protein undergoes N-linked glycosylation [20,21]. The glycosylation plays vital role in organ tropism and IFN signalling [22].

Envelope (E) protein is smallest amongst all the structural proteins, around 8–12 kDa, and plays major role in pathogenesis, virus assembly, and release [23]. Using solid-state NMR spectroscopy, one study demonstrated the structure and drug binding of SARS-CoV-2 E protein. The E protein topology of SARS-CoV-2 represented a five-helix bundle surrounding adehydrated narrow pore with bipartite channel. Although, E proteins are highly divergent in terms of amino acid composition, but structurally preserved across various genera of β-coronaviruses with a short hydrophilic N-terminus, a large hydrophobic region, followed by hydrophilic C-terminal tail [24].

The N (nucleocapsid) protein is solely complexed in structural organization of the nucleocapsid. It distinctly possesses three highly conserved domains; an N-terminal domain, an RNA-binding domain or a linker region, and a C-terminal domain [25]. It has been observed that these three domains may together orchestrate RNA binding [26], and its phosphorylated status is prerequisite for triggering a structural dynamism facilitating the affinity for viral versus non-viral RNA [27]. N protein participates in RNA packaging in a beads-on-a-string type conformation. In addition to be involved in organization of viral genome, N protein also facilitates virion assembly and enhances virus transcription efficiency amongst others [26,28]. Owing to considerably high immunogenic nature, N protein may be useful as potential vaccine target. Most importantly, the M, E, and S proteins possess trafficking signal sequence, which enable their translocation to the endoplasmic reticulum (ER).

## 5. Description of Non-Structural Proteins

Apart from aforementioned structural proteins there are several non-structural proteins, namely NSP1 to NSP 10 and NSP12 to NSP16, encoded by genes located within the 5′-region of viral RNA genome [29]. These non-structural proteins with their corresponding functions along with other associated molecular features are tabulated below (Table 1).

## 6. Description of Accessory Factors

There are nine accessory proteins—ORF3a, 3d, 6, 7a, 7b, 8, 9b, 14, and 10—produced from at least five ORFs encoding accessory genes (ORF3a, ORF6, ORF7a, ORF7b, and ORF8), novel overlapping ORF3d (earlier known as 3b), leaky scanning of sgRNA of N gene (ORF9b and 14), and ORF10 from downstream of N gene (Figure 1). Aforementioned accessory genes show considerable variability amongst coronavirus group. However, number of exact ORFs is still debated and awaits further experimental verification. These proteins along with abovementioned NSPs play very crucial role in viral replication.

### 6.1. ORF3a and ORF3d Proteins

The accessory factor 3a is encoded by ORF3a located in between the S and E genes, and is the largest accessory proteins of SARS-CoV-2, consisting of 274 amino acid residues. Hydrophobicity analysis and topology studies have revealed that 3a protein is an O-linked glycosylated, possessing three transmembrane domains. ORF3a forms dimer and its six transmembrane helices together create ion channelin the host cell membrane, which is highly conducive for Ca^2+^/K^+^ cations compared to Na^+^ ion [40]. It is also involved in virus release, apoptosis and pathogenesis [41]. Similarly, ORF3d (one amongst youngest genes) encodes 3d protein which consists of 154-aa long polypeptide chain, and is found to be located in the nucleolus and mitochondria. However, the presence of ORF3d in SARS-CoV-2 has also been confirmed [12].

### 6.2. ORF6 Protein

SARS-CoV ORF6 protein is a 61-amino acid long membrane-associated protein. The expression of this protein was confirmed in virus-infected Vero E6 cells, and also in the lung and intestine tissues of patients. It is actually placed in the ER and Golgi compartments in expressing cells and virus infected cells [42].

### 6.3. ORF7a and ORF7b Proteins

ORF7a and ORF7b accessory proteins are synthesized from the bicistronic subgenomic RNA 7 of SARS-CoV-2. The 122-aa ORF7aprotein is a type-I transmembrane protein containing a 15 aa signal peptide sequence, an 81aa luminal domain, 21aa transmembrane domain and a short C-terminal tail [43,44].

The ORF7b protein consists of 44-amino acids, and is an integral membrane protein, expressed in SARS-CoV-infected cells wherein it remains localized in the Golgi compartment. Furthermore, there is production of anti-7b antibody in SARS patient serum, indicative of its expression in infected patients. In addition, 7b protein was found to be closely associated with intracellular virus particles, strengthening the findings related to its expression and importance [43,44].

### 6.4. ORF8 Protein

ORF8, one amongst youngest genes, shows low homology to SARS-CoV due to deletion. This protein consists of 121 amino acid residues, and its shape resemble immunoglobulin (Ig)-like fold owing to β-strand core (18–121 residues). The 1–17 residues comprise N-terminal siganal sequence, prerequisite for transport to ER. ORF8 has been found to interact with major histocompatibility complex-I (MHC-I), thereby mediating their degradation in cell culture, and therefore may help in immune evasion [45].

### 6.5. ORF9b Protein

It consists of 97 amino acid residues, and is probably expressed by leaky scanning of sgRNA of N gene. It tends to associate with adaptor protein, TOM70, and therby suppress IFN-I mediated antiviral response [46]. Therefore, developing greater insight into 9b-TOM70 interaction may help in desiging therapeutic molecule.

### 6.6. ORF10 Protein

Gene encoding ORF10 protein is predicted to be located downstream of the N gene. Although its corresponding sgRNA is rarely detected, however ORF 10 protein has been found in infected cells [47].

### 6.7. ORF14 Protein

It is made up of 73 amino acid residues, and is also likely to be synthesized by leaky scanning of sgRNA of N gene [48]. However, its function is not clearly understood.

Apart from crucial role in virus replication, accessory proteins may also be involved in host immune escape. For instance, during infection of MERS-CoV, accessory ORFs 3–5 inhibit the host’s innate immune response, including perturbation of type I interferon, blockade of NF-κB and RNaseL activation, among others [49,50,51]. Study from Li J.Y. et al. has demonstrated the role of ORF6, ORF8, and nucleocapsid protein in inhibiting type I interferon (IFN-β) and NF-κB-responsive promoter, and impede interferon signaling [52]. Following study from Miorin et al. demonstrates the impairment in nuclear translocation of STAT1 and STAT2 leading to inhibition of transcriptional induction of IFN-stimulated genes [53]. Another report has shown the association of SARS-CoV-2 ORF9b to host mitochondrial import receptor subunit (TOM70), and thereby suppresses type I interferon signaling [46].

## 7. Life Cycle of SARS-CoV-2

Life cycle of SARS-CoV-2 consists of cellular invasion of virus, expression of viral genes, and formation of progeny and eventual exit. It can roughly be divided into following 5 steps (Figure 2).

### 7.1. Attachment to Host Cell Surface

The S protein system is homotrimeric, consisting of three monomeric S polypeptides. Each monomeric polypeptide contains S1 and S2 subunits with multiple functional domains and motifs, which undergo several conformational changes. SARS-CoV-2 via its receptor binding domain (RBD)of S1 subunit (RBD/S1 with down-to-up conformation) of homotrimeric spike glycoprotein binds host cell receptor ACE2 (angiotensin-converting enzyme 2) [14] zinc-binding carboxypeptidase, which is normally involved in cardiac function and blood pressure regulation. ACE2 is primarily expressed by epithelial cells of the lungs and small intestine as well as kidney, heart, and other tissues [54,55]. In addition, a recent molecular simulation-based work has proposed that SARS-CoV-2′s S protein can also bind to nicotinic acetylcholine receptors (nAChRs), indicative of its diverse binding potential, and may be one of the underlying reasons for multi-organ pathogenesis [56]. Whereas the S2 domain of S protein possesses heptad repeat region and fusion peptide, mediating fusion following conformational rearrangement [57]. Following the binding of S protein of SARS-CoV-2 to the host protein ACE2, the spike protein undergoes protease-mediated cleavage at the S1/S2 cleavage site for priming and a cleavage for activation at the S′2 site, a position adjacent to a fusion peptide within the S2 subunit. The S1/S2 site are also subjected to cleavage by other proteases such as transmembrane protease serine, 2 (TMPRSS2), cathepsin L and/or other proteases. Subsequent cleavage at the S′2 site presumably triggers membrane fusion via irreversible, conformational changes and thereby facilitates access to host cell cytosol [58,59]. In addition to the involvement in infection and cross-species transmission, the S protein is crucial target for anti-virus neutralizing antibodies, and crucial mutation in it may lead to considerable alteration in pathogenesis.

### 7.2. Viral Penetration and Uncoating

After fusion of viral spike glycoprotein with ACE2 there is subtle conformational changes, releasing the viral nucleocapsid into the cell cytosol. This process is aided by several host factors, including type II transmembrane protease serine 2 (TMPRSS2) protease and cathepsin L.

### 7.3. Replication-Transcription Complex (RTC) Formation

Immediately after release of viral nucleocapsid, +ssRNA serves as functional mRNA with respect to ORF1a and ORF1b encoding polyprotein pp1a (440–500 kDa) and pp1ab (740–810 kDa), respectively. However, pp1a is 1.2–2.2 folds more expressed compared to pp1ab owing to differential efficiency of frameshift between ORF1a and ORF1b genes [60]. These two polyproteins undergo autoproteolytic processing yielding 16 nsps, which together form the RTC for viral RNA synthesis. This functional RTC results into formation of a nested set of sgRNAs via discontinuous transcription [61].

### 7.4. Synthesis of Viral RNA

The formation of RTC sets molecular process in motion leading to synthesis of multiple copies of viral RNA. These -ssRNA (negative ssRNA) serves as intermediate template. Meanwhile, polymerase switches template at short motifs, transcription regulated sequences (TRS) during -ssRNA synthesis, thereby producing a multiple 5′-nested set of negative sense sgRNAs which, in turn, used as templates to form a 3′-nested set of positive sense sgRNAs. Thereafter, they associate with host ribosome, synthesizing various structural and accessory proteins building multiple virus structure [62].

### 7.5. Molecular Assembly and Release of SARS-CoV-2

Most of the structural and accessory proteins associated with membrane such as S, M, and E are synthesized by endoplasmic reticulum-bound ribosomes, whereas other viral proteins, including N protein, are translated by free cytosolic ribosomes of host cells. In addition, these structural proteins also undergo posttranslational modification that modulate their functions. The assembly of virion converges at site of endoplasmic reticulum–Golgi intermediate compartment (ERGIC), wherein M protein provides scaffold and orchestrate virion morphogenesis by heterotypic interaction with other structural proteins, such as M-S and M-E, thereby facilitating molecular incorporation. Furthermore, M-N interactions mediates condensation of the nucleocapsid with the envelope along with E protein [61]. Post molecular assembly, progeny virions are transported in smooth-wall vesicle and using secretory pathway they are trafficked to plasma membrane and eventually exit though exocytosis and spread to other parts of body [63,64].

## 8. Potential Therapeutic Targets for Drug Designing Against COVID-19

Currently, SARS-CoV-2 has rapidly spread across world beginning from Wuhan, capital of Central China’s Hubei province, causing a fatal outbreak of acute infectious pneumonia. Till date, considerable success has not been achieved regarding development of specific anti-virus drugs or vaccines for the treatment of this disease despite consistent efforts made by researchers around the world. Nonethless, lots of efforts are being made to drug repurposing of FDA-approved/clinical-trial drugs in order to use them against COVID-19. In this direction, various pharmaceutical companies and government agencies have reportedly succeeded to a certain extent, necessitating further research work to find the cure. There are various potential therapeutic drug molecules, compounds and antibodies under clinical trial, offering considerable benefits are listed in Table 2 below.

## 9. Plasma Therapy and Neutralizing Antibody Cocktail

Plasma therapy is being tested and adopted as one of the crucial therapeutic regimes to treat COVID-19 patients. This is based on fact that polyclonal antibodies produced in convalescent persons following infection may help neutralize viruses, and substantially reduces the duration of viremia and hospitalization. During this therapy, isolated anti-SARS-CoV-2 antibodies from recovered patients are administered in patient in order to effectively neutralize the virus through multiple immunological mechanisms. Besides, it could also provide prophylactic immunity prior to the occurrence of viral infection. Furthermore, various epitopes on the spike protein can be targeted using antibody cocktails. This line of supportive treatment has shown quite promising results in preliminary investigations carried out across the world, and therefore, is being scaled up to an unprecedented level [75,76]. However, it requires further detailed investigation and optimization at multiple levels, including antibody concentration, dose and frequency of administration. Besides, genetic engineering has enabled development of single-domain antibodies (sdAbs), or nanobodies against S and N proteins in order to prevent viral attachment to host cells by blocking ACE2 binding [77]. One of the central hypotheses regrading coronavirus disease is based on the empirical observation that complications and death are the consequences of viral load whose reversal may bring clinical benefits to the patient concerned. Following this hypothesis, a recent study involving RGEN-COV2 (antibody cocktail consisting of two fully human antibody, noncompeting IgG1 against RBD of SARS-CoV-2 S protein) has shown substantial benefit in term of enhanced viral clearance and hence, may be adopted as antiviral therapy, particularly in patient with weakened immune response [78]. Cytokine storm following SARS-CoV-2 infection leads to substantial complications and tissue damage leading to ARDS (Acute Respiratory Distress Syndrome). Therefore, controlling cytokine storm by using therapeutic antibodies, which can target TLR4 (EB05), CXCL10 (EB06), and IL6 (Levilimab) amongst others, may suppress proinflammatory response and hence consequent histological damage in multiple tissues following infection [75,79].

## 10. Vaccines for COVID-19

The worldwide endeavor for manufacturing vaccines to tackle SARS-CoV-2 infection has been growing since the beginning of the current pandemic. Vaccine is administered to all age groups of people to help them develop and strengthen both humoral and cell-mediated immunity so that they can fight infection. Until now, there have been the development and authorization/approval of 13 vaccines worldwide, and around 58 vaccines are under various phases of trial (https://www.raps.org/news-and-articles/news-articles/2020/3/covid-19-vaccine-tracker; accessed on 30 March 2021). Inactivated vaccine accounts for 38% of total approved vaccines worldwide (Figure 3). The various aspects of 13 approved vaccines are mentioned in Table 3.

## 11. Conclusions

The SARS-CoV-2 is the causative pathogen for the current pandemic, and is evolving through recombination and mutation into several strains over a period of time. Owing to free geopolitical borders, lack of knowledge regarding its spread, and initial negligence by various stakeholders have led to quick spread of COVID-19, causing millions of deaths and debilitation, as well as huge burden on socio-economic and health system of nations, and territories worldwide. Nevertheless, technological and methodological advances in the field of virology, molecular biology, and pharmacology haves helped us understand the structural and genomic organization as well as mechanism underlying cellular entry, lifecycle, and pathophysiological characteristics of SARS-CoV-2 to some extent. However, there is still need for establishing cellular and animal models for SARS-CoV-2 to develop even greater insights into mechanisms underlying viral replication, pathogenesis, and transmission dynamics. Globally, such studies are being carried out, aiming at the development of therapeutic strategies against the zoonotic coronavirus epidemic. The development of therapeutic strategies could rely upon studying the molecular mechanisms underlying host–pathogen interaction, ever-evolving virus genome through recombination and mutation, host immune response involving innate and adaptive immunity, and so forth. Amongst such therapeutic studies are the development and/or repurposing of various drugs (Table 2), targeting cellular entry and replication to reduce the severity of the disease. In addition, plasm therapy is also being used as a supportive treatment to neutralize the virus and as a prophylactic measure as it involves mixture of antibodies against multiple epitopes on various structural moleculessuch as S and N proteins. Genetically engineered nanobodies and antibodies suppressing cytokine storms have also shown promising results [77,79]. The worldwide effort has also resulted in the development of various types of vaccines, which are being administered, and promising results have been obtained. Therefore, considering above advancement in the field of etiology, drug, and vaccine development, we may hope for promising outcome in near future.

## Figures and Tables

**Figure 1 cells-10-00821-f001:**
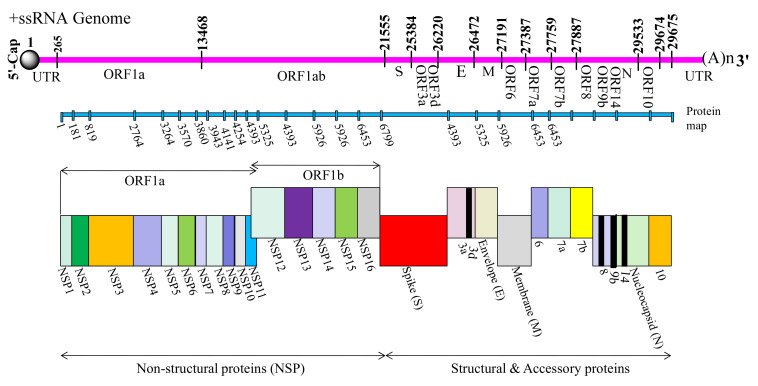
Genomic arrangement of SARS-CoV-2. The genomic organization of the SARS-CoV-2 shows sequential arrangement of various non-structural, structural and accessory genes as follow: 5′-cap-leader-UTR-replicase-S (Spike)–E (Envelope)-M (Membrane)-N (Nucleocapsid)-3′UTR-poly (A) tail with accessory genes such as 3a, 3d, 6, 7a,7b, 8, 9b, 14, and 10 interspersed among the structural genes preceding 3′ end of the viral RNA genome.

**Figure 2 cells-10-00821-f002:**
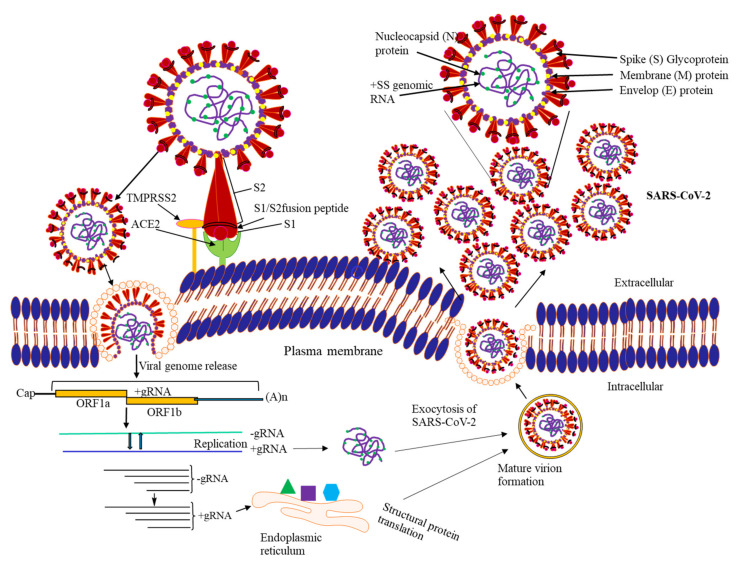
SARS-CoV-2 begins by binding with its S protein (RBD/S1) on host cell receptor, ACE2, driving conformational change in the S2 subunit, and thereby facilitating its fusion with plasma membrane. Immediately after release of +ssRNA, translation leads to formation of non-structural polyproteins pp1a and pp1ab, which undergo proteolytic cleavage and are eventually assembled into functional replicase. The replicase leads to formation of a negative-sense intermediate, which is eventually replicated to form multiple copies of gRNA as well as nested set of sgRNA by discontinuous transcription. These sgRNA are translated into various structural and accessory proteins, which are assembled as virion in the ERGIC, and eventually exit cell via exocytosis.

**Figure 3 cells-10-00821-f003:**
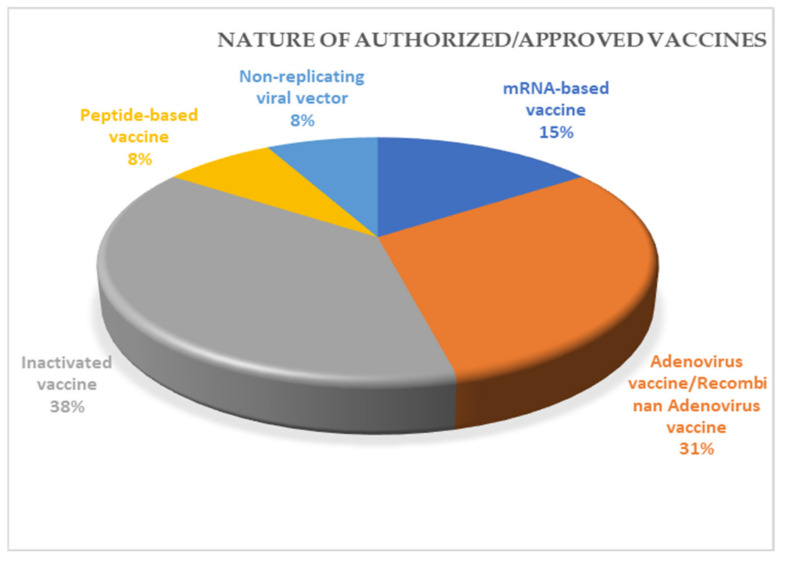
Nature of various types of authorized vaccines worldwide.

**Table 1 cells-10-00821-t001:** Brief description of various non-structural proteins of SARS-CoV-2.

Name	Protein (Full Name)	Length (aa)	Range	Role	Accession No.
NSP1	N-terminal product of the viral replicase	180	1–180	Leader protein which acts as host translation inhibitor and also degrade host mRNAs [30]	YP_009725297
NSP2	N-terminal product	638	181–818	Binds to prohibitin 1 and prohibitin 2 (PHB1 and PHB2) [31]	YP_009725298
NSP3	Papain-like proteinase	1945	819–2763	Responsible for release of NSP1, NSP2, and NSP3 from the N-terminal region of pp1a and 1ab [32]	YP_009725299
NSP4	Membrane-spanning protein containing transmembrane domain 2	500	2764–3263	Viral replication-transcription complex and it helps modify ER Membranes [33]	YP_009725300
NSP5	Proteinase and main proteinase	306	3264–3569	Cleaves at multiple distinct sites to yield mature and intermediate nonstructural proteins	YP_009725301
NSP6	Putative transmembrane domain	290	3570–3859	Induces formation of ER-derived autophagosomes As well as induces double-membrane vesicles [34]	YP_009725302
NSP7	RNA-dependent RNA polymerase	83	3860–3942	Forms complex with NSP8 and NSP12 to yield the RNA polymerase activity of NSP8 [35]	YP_009725303
NSP8	Multimeric RNA polymerase; replicase	198	3943–4140	Makes heterodimer with NSP8 and 12 [36]	YP_009725304
NSP9	single-stranded RNA-binding viral protein	198	4141–4253	May bind to helicase	YP_009725305
NSP10	Growth-factor-like protein possessing two zinc binding motifs	139	4254–4392	Yet to be deciphered	YP_009725306
NSP11	Consists of 13 amino acids (sadaqsflngfav) andidentical to the first segment of Nsp12	13	4393–4405	Unknown	YP_009725312
NSP12	RNA-dependent RNA polymerase	932	4393–5324	Replication and methylation [37]	YP_009725307
NSP13	RNA-dependent RNA polymerase (Pol/RdRp)	932	5325–5925	A helicase core domain that binds ATP. Zinc-binding domain is involved in replication and transcription [38]	YP_009725307
NSP14	Proofreading Exoribonuclease domain (ExoN/nsp14)	527	5926–6452	Exoribonuclease activity acting in a 3′-5′ direction and N^7^-guanine methyltransferase activity	YP_009725309
NSP15	EndoRNAse; nsp15-A1 and nsp15B-NendoU	346	6453–6798	Mn(2 + )-dependent endoribonuclease activity	YP_009725310.1
NSP16	2′-O-ribose methyltransferase	298	6799–7096	Methyltransferase that mediates mRNA cap 2′-O-ribose methylation to the 5′-cap structure of viral mRNAs [39]	YP_009725311

**Table 2 cells-10-00821-t002:** Selected list of therapeutic molecules currently in clinical trial and their respective targets on SARS-CoV-2.

S/No	Drug/Compounds/Antibody	Potential Targets Available on SARS-CoV-2
1	Pyridone-containing α-ketoamides	Targets M^pro^, also referred to as 3C-like proteinase or NSP5, and thereby interfering with viral replication [65]
2	Chloroquine and formoterol	They may act as papain-like protease (PLpro), inhibiting proteolytic generation and maturation of NSP1, NSP2 and NSP3 thereby interfering with virus replication [66]. Chloroquine interferes with terminal glycosylation of ACE2 receptor, thereby inhibiting its interaction with S protein of SARS-CoV-2
3	Remdesivir (nucleotide analog)	It gets incorporated in nascent viral RNA, and thereby inhibiting the RdRp and hence it may be very effective against COVID-19 [67]
4	β-D-N4-hydroxycytidine (ribonucleoside analog)	Inhibits viral replication [68]
5	Bananin (adamantane derivative)	Inhibits helicase NSP13 and therefore may prevent viral replication [69]
6	Camostat mesylate & Bromhexine hydrochloride	Prevents TMPRSS2-mediated viral entry [70], and acts as TMPRSS2 inhibitor [70], respectively
7	Zidovudine	May play antiviral by binding with nucleocapsid (N) phosphoprotein protein [71]
8	CR3022 (monoclonal antibody)	Binds RBD of S protein and therefore may prevent cellular interaction of virus [72]
9	Ivermectin	Ivermectin binds to and destabilises nuclear transporter, Impα/β1 heterodimer, preventing its binding to the viral cargo proteins and their translocation into the nucleus. This prevents cargo mediated- suppression of antiviral response and therefore, reduces viral load by ~5000 folds [73]
10	Ebselen	Reduce COVID-19 by 20.3 folds [74]

**Table 3 cells-10-00821-t003:** List of authorized/approved vaccines against SARS-CoV-2 for COVID-19 (^$^ FDA Approved, Emergency Use Authorization (EUA) vaccines).

S/No	Vaccine Name	Vaccine Type	Developers	Country of Origin	Current Schedule and Route of Administration	Reported Effectiveness Following Clinical Trial
1.	Comirnaty (formerly BNT162b2) ^$^	mRNA-based vaccine(encodes mutated form of S protein)	Pfizer, BioNTech; Fosun Pharma	Multinational	Two doses, 21 days apart, intramuscular injection	95% efficacy in Phase 3 clinical trial (NCT04368728) [80]. 92% efficacy in vaccinated healthcare workers [81].
2	Moderna COVID-19 Vaccine ^$^ (mRNA-1273)	mRNA-based vaccine	Moderna, BARDA, NIAID	USA	Two doses, 28 days apart, intramuscular injection	94.1% efficacy in Phase 3 clinical trial (NCT04470427) [82].
3	COVID-19 Vaccine Janssen (JNJ-78436735; Ad26. COV2.S) ^$^	Non-replicating viral vector	Janssen vaccines (Johnsons & Johnsons)	The Netherlands, US	Single dose vaccine, intramuscular injection	85% efficacy in Phase 3 ENSEMBLE trial (NCT04505722).
4	COVID-19 Vaccine AstraZeneca (Covishield)	Adenovirus vaccine	BARDA, OWS	UK	Two doses, between 4–12 weeks apart, intramuscular injection	79% efficacy in Phase 3 clinical trial (NCT04516746). 100% efficacy in severe disease and hospitalization patients.
5	Sputnik V (Gam-COVID-Vac)	Recombinant adenovirus vaccine (rAd26 and rAd5)	Gamaleya Research Institute, Acellena Contract Drug Research and development	Russia	Two doses, 21 days apart, intramuscular injection	94.1% efficacy in Phase 3 clinical trial (NCT04530396) [83].
6	CoronaVac (formerly PiCoVacc)	Inactivated vaccine (formalin with alum adjuvant)	Sinovac	China	Two doses, between 14–18 days apart, intramuscular	50% efficacy in Phase 3 clinical trial (NCT04456595) [84].
7	BBIBP-CorV	Inactivated SARS-CoV-2 vaccine (Vero cell)	Beijing Institute of Biological Products; China National Pharmaceutical Group (Sinopharm)	China	Two doses, intramuscular injection	86% efficiency Phase 3 clinical trial (ChiCTR2000034780). High effectiveness in terms of neutralizing antibody production in rhesus macaques.
8	EpiVacCorona	Peptide vaccine	Federal Budgetary Research Institution State Research Center of Virology and Biotechnology	Russia	Two doses, 21–28 days apart, intramuscular injection	Phase1/2 trial (NCT04527575) Trial is still going and evaluation regarding efficiency being carried out.
9	Convidicea (Ad5-nCoV)	Recombinant vaccine (adenovirus type 5 vector)	CanSino Biologics	China	Single dose vaccine, but also evaluated in trial with 2-doses, intramuscular	65.7% efficiency in Phase 3 clinical trial (NCT04526990).
10	Covaxin	Inactivated vaccine	Bharat Biotech in collaboration with National Institute of Virology), ICMR.	India	Two doses, intradermally	81% in Interim phase 3 trial [74].
11	Name is yet to be specified	Inactivated vaccine	Sinopharm and the Wuhan Institute of Virology under the Chinese Academy of Sciences	China	Final number of doses and interval not yet decided	Phase1/2 clinical trial (ChiCTR2000031809) is completed and 72.51% efficacy in on-going phase 3 clinical trial.
12	CoviVac	Inactivated vaccine	Chumakov Federal Scientific Center for Research and Development of immune and Biological Products	Russia	Not yet finally decided	Phase1/2 trial is undergoing.
13	ZF2001	Recombinant vaccine (CHO)	Anhui Zhifei Longcom Biopharma ceutical, Institute of Microbiology of the Chinese Academy of Sciences	China, Uzbekistan	Not yet finally decided, intramuscular injection	Phase 3 clinical trial (NCT04646590) is being evaluated.

## Data Availability

Not applicable.

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
