# Peer review of "Role of Structural and Non-Structural Proteins and Therapeutic Targets of SARS-CoV-2 for COVID-19"

_cells, 2021, doi:10.3390/cells10040821_

Round 1

Reviewer 1 Report

General appreciation:

The review is clear and globally well written. It will be interesting to read for people working on SARS-CoV-2 and get an overview on the different proteins from the virus and their role.

However, the part on accessory proteins contains some imprecision and lacks important references.

In addition the last part on therapeutics targets is quite short and would deserve a more substantial text and table.

Please find below detailed comments on the different parts of the manuscript:

Introduction:

- It would be more relevant to give approximate number of cases and deaths due to COVID rather than precise numbers, since we know these numbers are only an estimation and because they evolve every days very quickly. For example, the “93536554” reported cases could be replaced by “nearly 100 millions” (or, to the day of the review: nearly 120 millions).

Description of structural proteins

- The authors say “E protein with unresolved topology till date”, but some information is known from the solid-state NMR structure of the transmembrane domain of E. Reference:

Mandala, V. S., McKay, M. J., Shcherbakov, A. A., Dregni, A. J., Kolocouris, A., & Hong, M. (2020). Structure and drug binding of the SARS-CoV-2 envelope protein transmembrane domain in lipid bilayers. Nature Structural & Molecular Biology, 1–24. http://doi.org/10.1038/s41594-020-00536-8

Genomic arrangement of SARS-CoV-2

- page 3: authors list only 6 accessory proteins: ORF3a, ORF6, ORF7a, ORF7b and ORF8. Yet, the authors have ORF10 in their figure 1, but not in the text. And later in the text they say there are 8 accessory proteins (page 5). If the exact number of accessory genes of SARS-CoV-2 is still debated which can be confusing, at least authors should be consistent within their own text.

As a consensus, in the most recent literature, there are commonly 9 accessory proteins described. The missing proteins should therefore be added to the list: ORF9b, ORF10, ORF14 and ORF3d, which was previously and wrongly called ORF3b first.

Reference for ORF3d discovery and replacement of ORF3b:

Nelson, C. W., Ardern, Z., Goldberg, T. L., Meng, C., Kuo, C.-H., Ludwig, C., et al. (2020). Dynamically evolving novel overlapping gene as a factor in the SARS-CoV-2 pandemic. eLife, 9. http://doi.org/10.7554/eLife.59633

- Please redo Figure 1 including ORF3d, ORF9b and ORF14.

Description of accessory factors:

- Here again, there is a confusion in the accessory proteins. Authors say “There are eight accessory proteins, viz., 3a, 3b, p6, 7a, 7b, 8b, 9b, and ORF14.” Please replace by “There are nine accessory proteins: ORF3a, 3d, 6, 7a, 7b, 8, 9b, 10 and 14.”
Be careful with the naming: ORF8b only exists in SARS-CoV-1, but in SARS-CoV-2 it is called ORF8 and not ORF8b.

- Confusing sentences: “These proteins are dispensable for viral replication in vitro; however, they might play important role in vivo. NSPs and aforementioned accessory factors play very crucial role in viral replication 36. “ It is not clear from these 2 sentences if accessory proteins are dispensable or crucial in viral replication? Reference n° 36 does not enable to state about this. From my knowledge, some accessory proteins are rather involved in the inhibition of type I interferon signaling pathway (see references above).

- Authors say “However, the presence of ORF3b in SARS-CoV-2 is highly speculative. “ Indeed, as explained above, a recent study has shown that this protein should be replaced by the smaller ORF3d. The paragraph on ORF3b could be modified accordingly.

- orf6 is 61 aa, not 63. Please replace.

- orf6: missing reference(s) for the following statement: “The expression of this protein was confirmed in virus-infected Vero E6 cells and also in the lung and intestine tissues of patients. It is actually placed in the ER and Golgi compartments in expressing cells and virus infected cell. ”

- Same remark for orf7a/b: please include reference(s).

- page 6: the authors mention the perturbation of type I interferon pathway by accessory proteins in the MERS-CoV. However they forget to mention more recent papers showing important results on SARS-CoV-2. Below are at least 3 examples on ORF6, ORF8 and ORF9b. The authors should add or replace this paragraph with what is known to date on SARS-CoV-2 accessory proteins regarding interferon signaling.

 Li, J.-Y., Liao, C.-H., Wang, Q., Tan, Y.-J., Luo, R., Qiu, Y., & Ge, X.-Y. (2020). The ORF6, ORF8 and nucleocapsid proteins of SARS-CoV-2 inhibit type I interferon signaling pathway. Virus Research, 286, 198074. http://doi.org/10.1016/j.virusres.2020.198074

Miorin, L., Kehrer, T., Sanchez-Aparicio, M. T., Zhang, K., Cohen, P., Patel, R. S., et al. (2020). SARS-CoV-2 Orf6 hijacks Nup98 to block STAT nuclear import and antagonize interferon signaling. Proceedings of the National Academy of Sciences of the United States of America, 117(45), 28344–28354. http://doi.org/10.1073/pnas.2016650117

Jiang, H.-W., Zhang, H.-N., Meng, Q.-F., Xie, J., Li, Y., Chen, H., et al. (2020). SARS-CoV-2 Orf9b suppresses type I interferon responses by targeting TOM70. Cellular & Molecular Immunology, 17(9), 998–1000. http://doi.org/10.1038/s41423-020-0514-8

Life cycle:

-This paragraph is clear and well-illustrated with Figure 2. Figure 2 would yet be clearer if the virion entry would be drawn on the left rather than on the right, which is an unusual representation of virus life cycle in general.

-Maybe authors could add a few more references to support their statements, especially in the last paragraph “Molecular assembly and release of SARS-CoV-2 ”.

Potential therapeutic targets for drug designing against COVID-19

The paragraph on therapeutic targets seems a bit ‘light’. When reading the title of the review, readers will certainly expect a more detailed part on therapeutic targets.

How did the authors choose the few drugs to appear in Table 2?

I understand that it is certainly not possible to list all the drugs/antibodies currently in clinical trial, yet it would be nice to have a more complete overview of the therapeutic targets with more examples. I suggest that the authors write a more substantial paragraph on this topic, otherwise the term ‘therapeutic targets’ should be removed from the title.

English and formatting – minor corrections:

- The Latin abbreviation ‘viz.’ is not common. I recommend the authors to replace it by ‘namely’ in the many occurrences.

- page2: many lines are written in italic at the beginning of the page. Please remove italic when not applicable.

- page 3: sub-units: replace by subunits

- abbreviations should only be used the first time. For example, Spike protein is re-abbreviated 5 times. Furthermore, some abbreviations are never used again (ex: NTD, CTD).

- page5: only subtitles should be shown in bold, not the whole paragraph.

-Table 1 pages 4-5: remove all ‘It’ from the “Role” column to be consistent

-page 5:  replace ‘may binds’ by ‘may bind’

- page 7: replace china by China

- page 7: There is no legend for Table 2.

Author Response

Please find the attached file for reviewer -1 comments

Reviewer 2 Report

The review study by Chaudhary et al. related a description of structured and non-structured proteins in SARS-CoV-2 and shortly highlight the stage of therapeutics. The authors need to address the following comments:

  1) Abstract. I prefer the author replace the word "molecular slicing" which is not suitable in in this context and replace by cleavage of the Spike (S) protein. Review the whole abstract for typos and grammar errors.   2)Page 2, instead of "93536554 reported cases" the construction "over 100M (now 118M) confirmed cases" suits here. We know due to the high spreading of the diseases the exact numbers  change from day to day. The same for death toll. Also, update this numbers respect to current situation.

  3) Page 2, replace dreaded by deadly   4)In general improve the intro, it is quite repetitive.    5) In the section Genomic... : The range 29.8 kb to 29.9 kb is odd (what is in 0.1 kb). Revise the message.   6) Page 3, in Description of Structural proteins. The author confuse the molecular structure of the S protein with two domain S1 and S2. The aim of the section is a bit lost and lack some reference to emphasise the message. For instance, there is pay attention to the role of the S protein during cell entry and its stability as the major role. Please  see  https://doi.org/10.3390/ma13235362 and https://doi.org/10.1039/D0NR03969A studies.   7) Page 4. A reference is needed by the end of the first paragraph: ...(ER) stress response.   8) As a general comment avoid redundancies and frequent use of words such as "viz". They can take namely, i.e., etc   9) Page 6, I sectionLife Cycle of SARS-CoV-2. Review it for repetitions and also improve the message in : one process leads to another in a synchronized manner, releasing the viral genome into cytosol. What process, what leads to.. Clear example of vague description.   10) Page 7, define acronym ERGIC   11) Page 7, Figure 2 The entry process is not clear, replace S1---> RBD/S1, as the RBD bind the ACE2 receptor. Also, the scale of spike protein binding the ACE2 is not conveniently represented. Perhaps a zoom into with an small side panel in order to enlarge the contact region will work better. In the abstract what is sgRNA?   12) Definitely, I expect the authors to modify to some extend the section regarding therapeutics. The are not key studies/references. The information is outdated No hints to current vaccines in place around the world, also not information about plasma therapy, cocktail of AB,  FDA-approved drugs, etc. See review https://doi.org/10.1080/07391102.2020.1758788    13) Conclusions have to change accordingly.

Author Response

Please find response letter of reviewer-2 comments as attached file. 

Round 2

Reviewer 2 Report

I see the authors have tried to implement partially some of my comments. However, they have left behind some of the concerns raised in my first evaluation which show the diligent work carry out by the author in this mini-review. I suggest to pay attention to those details for the sake of success of their work. Among them, I consider very relevant to clarify the structure of  trimeric S protein: one S1 and one S2 domain per chain. Authors insist to write: 

author text: The S protein of virion is a type I transmembrane N-linked glycosylated protein of around 150-200 kDa with two and one S2 unit making them trimeric molecule [13].

So far, the author confused the well-established molecular structure of the S protein with two domain S1 and S2. The aim of the section is a bit lost and still lacks some references. Such as the role of the S protein during cell entry and its stability as the major role. Please  see  https://doi.org/10.3390/ma13235362 and https://doi.org/10.1039/D0NR03969A studies.  And, I guess the author should have also added more details about the glycosylation state of the S protein and its relevance in the prefusion state (doi.org/10.1021/acscentsci.0c01056).

A very urgent part is to extend/correct the section regarding therapeutics. There are not key studies/references. The information is totally outdated (see below):

Authors test: Till date, there have not been success regarding development of specific anti-virus drugs or vaccines for the treatment of this disease despite consistent efforts made by researchers around the world.

No hints to current vaccines (Pfizer, Moderna, Sputnik V, Astrazenca, etc) in place around the world, also not information about cocktail of antibodies,  FDA-approved drugs, etc. See review for it https://doi.org/10.1080/07391102.2020.1758788. 

Author Response

Dear Reviewer, We have addressed all of the comments raised in revised manuscript. 
